# Caregivers' mental distress and child health during the COVID-19 outbreak in Japan

**Sayaka Horiuchi**[1]*, **Ryoji Shinohara**[1], **Sanae Otawa**[1], **Yuka Akiyama**[2], **Tadao Ooka**[2], **Reiji Kojima**[2], **Hiroshi Yokomichi**[2], **Kunio Miyake**[2], **Zentaro Yamagata**[2]

**1** Center for Birth Cohort Studies, University of Yamanashi, Chuo-shi, Yamanashi, Japan, **2** Department of Health Sciences, School of Medicine, University of Yamanashi, Chuo-shi, Yamanashi, Japan

* sayakahoriuchi@gmail.com

**Data Availability Statement:** All relevant data are within the manuscript and its Supporting Information files.

**Funding:** The study was funded by the running expense of University of Yamanashi. The funder

## Abstract

To clarify the physical and mental conditions of children during the coronavirus disease 2019 pandemic and consequent social distancing in relation to the mental condition of their caregivers. This internet-based nationwide cross-sectional study was conducted between April 30 and May 13, 2020. The participants were 1,200 caregivers of children aged 3–14 years. Child health issues were categorized into "at least one" or "none" according to caregivers' perception. Caregivers' mental status was assessed using the Japanese version of the Kessler Psychological Distress Scale-6. The association between caregivers' mental status and child health issues was analyzed using logistic regression models. Among the participants, 289 (24.1%) had moderate and 352 (29.3%) had severe mental distress and 69.8% of children in their care had health issues. The number of caregivers with mental distress was more than double that reported during the 2016 national survey. After adjusting for covariates, child health issues increased among caregivers with moderate mental distress (odds ratio 2.24, 95% confidence interval 1.59–3.16) and severe mental distress (odds ratio 3.05, 95% confidence interval 2.17–4.29) compared with caregivers with no mental distress. The results highlight parents' psychological stress during the pandemic, suggesting the need for adequate parenting support. However, our study did not consider risk factors of caregivers' mental distress such as socioeconomic background. There is an urgent need for further research to identify vulnerable populations and children's needs to develop sustainable social support programs for those affected by the outbreak.

## Introduction

There are various ways in which the coronavirus disease 2019 (COVID-19) pandemic and consequent lockdown and social distancing measures have placed strain on the physical and psychological health of children and adolescents [1]. Parental fear of visiting health facilities has led to a reduction in vaccination rates and necessary health care visits [1, 2]. School closures have resulted in isolation of children and families, increased their stress, and increased children's risk of maltreatment at home [1, 3, 4]. In this already unprecedented situation, parental mental health problems related to fear of infection, economic pressure, and rapid changes in lifestyle are a major cause of concern [5]. However, the social distancing measures

had no role in study design, data collection and analysis, decision to publish, or preparation of the manuscript.

**Competing interests:** The authors have declared that no competing interests exist.

in place make it difficult to identify at-risk children and families who may be in need of social support. Children are also missing out on opportunities to study, exercise, and communicate with their peers and teachers, which are essential for healthy development [4].

In Japan, the social isolation measures instituted in response to an increasing number of COVID-19 cases included school closure between March and June, 2020 [6]. On April 7, the government declared a state of emergency, which was extended to the entire nation on April 16 and ultimately lifted on May 25 [7, 8]. During this period, people were asked to avoid unnecessary gatherings and outings without legal restrictions. Public health services for children and childrearing families—such as child health checkups, childrearing consultations for parents, school counseling for children, and social support for children with developmental disorders or disabilities—were also canceled during this period.

Regardless of increasing global awareness of child health concerns related to the pandemic and social isolation, surveys that clarify the actual situation of children and their families under the state of emergency in Japan are yet to be conducted. Therefore, the present study aimed to clarify the physical and psychological conditions of children and caregivers during the COVID-19 pandemic. Furthermore, we examined the association between caregivers' mental status and the physical and psychological condition of children, which is particularly significant at a time when families were isolated and children spent most of their time at home.

## Materials and methods

This study was approved by the Ethics Review Board of the University of Yamanashi (approval number: 2259). Informed consent was obtained via online.

### Design and setting

This internet-based cross-sectional study was performed between April 30 and May 13, 2020. The Prime Minister requested school closure on February 29 and many schools and childcare facilities remained closed until the lifting of the state of emergency.

### Participants

The target population was caregivers of children aged 3–14 years who had voluntarily registered with the Nippon Research Center as monitors for web-based surveys in response to online affiliate advertising. Eligible individuals were asked to participate in the survey through the center and only those who answered screening questions and provided their consent on the website answered the entire questionnaire. Data collection continued until the sample size reached 1,200. The sample size was calculated to detect a 20% absolute difference in the percentage of children with any physical or mental health problems between caregivers with and without mental distress based on reports of a 40–50% prevalence of mental distress in the general population during the pandemic.

### Data collection

The internet-based questionnaire consisted of 35 questions related to participants and their children. The questionnaire was developed by the authors. No validation test was performed prior to the survey. However, the Japanese version of the Kessler Psychological Distress Scale-6 (K6), which was included in the questionnaire, has proven validity in the context of mental health screening [9]. The Cronbach's alpha coefficient of the K6 score in this study was 0.93.

Participant-related questions included job type, time spent with the child, COVID-19-related concerns, and mental health status. Mental health status was assessed using the

Japanese version of the K6 [9, 10]. The K6 has demonstrated excellent internal consistency and reliability [11] and is widely used in epidemiological studies [12]. The properties of the Japanese version are comparable to those of the original (the area under the receiver operating characteristic curve was 0.94 (95% confidence interval [CI] = 0.88 to 0.99) [9], and this tool has been used to screen for depression and anxiety disorders in the workplace and in the Comprehensive Survey of Living Conditions in Japan [13]. Questions about the child included whether the school/nursery was open, frequency of playing outside, screen time, and health condition as perceived by the participant.

Data collection was commissioned by the Nippon Research Center. If participants had more than one child, they were asked to restrict their answers to one child, who was randomly selected by the system.

**Exposure.** The main exposure was caregivers' mental status, assessed using the Japanese version of the K6 [9]. Based on previous studies conducted in Japan, the K6 score was categorized into three groups: 0–4 (no mental distress), 5–9 (moderate mental distress), and $\geq$ 10 (severe mental distress) [9, 14].

**Outcome.** The primary outcome was child health condition as perceived by the participant. Child health status was measured by asking participants whether their child had at least one problem related to sleep, appetite, physical and mental conditions, activity, or behavior at the time of the survey. Sleep issues included difficulty in sleeping or waking up and disturbance of sleep rhythm. Appetite included both increased or reduced appetite and change in body weight and body image. Physical and mental conditions included dullness, fatigue, lack of energy, pale appearance, headaches, stomach aches, dizziness, and nausea. Activity was measured in terms of whether a child was reluctant to go out or meet friends. Behavior included repetitive actions, use of violence, being distracted, reduced emotional reactions, incoherent conversation, and a higher frequency of talking to oneself. Participants were asked to choose all applicable conditions.

**Covariates.** The covariates were the caregiver's gender and age, child's gender and age, geographical area of residence, whether the nursery/school was open at the time of the survey, child's frequency of playing outside, child's screen time, and time spent with the child during the daytime by both caregiver and partner. The child's screen time included the use of any device for any purpose, including home schooling. The family's socioeconomic status has previously been reported to be a risk factor for child mental health problems [5, 15, 16], and the questionnaire therefore asked about the respondents' job type and their partners. We categorized the job type into employed/self-employed, part-time, and unemployed/housewife/student. However other potential risk factors for child mental health problems such as educational attainment [15, 16] and past medical history [16] were not included in the questionnaire, owing to the sensitivity of these topics and with a view to increasing the response rate.

## Data analysis

The distribution of each variable was analyzed and summarized as number (%) or mean (standard deviation) as appropriate. A univariate analysis was then performed to analyze the association between child health and the caregiver's mental status (K6 score) and each of the other covariates using logistic regression models. A multivariate analysis was performed to evaluate the association between the caregiver's mental status and child health by adjusting for all covariates in the logistic regression models. Finally, interactions between the effects of the caregiver's mental status and the caregiver's gender, caregiver's age, child's gender, child's age, and school closure on child health conditions were analyzed using likelihood ratio tests, as age and gender are important factors in determining children's vulnerability to environmental

change, such as the COVID-19 outbreak, according to previous studies [15, 16]. STATA/MP 16.1 software was used for all analyses.

## Results

The Nippon research center requested a total of 2792 people to answer the questionnaire. Among 1748 people who answered screening questions, 548 people were removed because they did not meet eligibility criteria (caregivers of children aged 3–14 years). Table 1 summarizes the characteristics of the participants. Among 1,200 participants, 289 (24.1%) had moderate mental distress (K6 score of 5–9) and 352 (29.3%) had severe mental distress (K6 score of $\geq$ 10). Approximately half of the participants were male (51.1%) and about half resided in a city with more than 300,000 people. Half of the children were male (52.7%), and the majority were school-aged (75.0%). At the time of the survey, the schools/nurseries of the children of 75.5% of the participants were closed, while in the case of 15.4% of the participants, the children had access to either online classes or special offline classes. There were no missing data.

Table 2 depicts the frequency of child health issues. The children of 69.8% of the participants had at least one health-related issue. The most frequent issue was change in sleep rhythm (57.3%), followed by change in appetite (28.4%).

Table 3 summarizes the crude and adjusted odds ratios (ORs) of child health issues in relation to participants' mental health status. The univariate analyses showed that moderate and severe mental distress in participants was associated with increased odds of their child experiencing health issues (moderate distress: OR 2.27, 95% CI 1.64–3.12, OR 2.86, 95% CI 2.09–3.91) compared with no mental distress. Furthermore, regarding the children, higher age, nursery/school being closed, lower frequency of playing outside, and increased screen time were associated with health issues. After adjusting for covariates, participants' mental status was still associated with child health issues. Odds of child health issues were higher among participants with moderate mental distress (OR 2.24, 95% CI 1.59–3.16) and severe mental distress (OR 3.05, 95% CI 2.17–4.29), compared with those with no mental distress. Increased screen time was also associated with child health issues in the adjusted model. There was no interaction between the effects of the caregiver's mental status and caregiver's gender ($p$ = 0.773), caregiver's age ($p$ = 0.110), child's gender ($p$ = 0.342), child's age ($p$ = 0.738), and school closure ($p$ = 0.437) on child health condition.

Caregivers' mental distress and increased screen time in children were associated with every aspect of child health issues: change in sleep, appetite, physical/mental condition, activity, and behavior (Table 4).

## Discussion

The present study revealed that the number of caregivers with moderate to severe mental distress during the state of emergency was more than double the number observed in national surveys, wherein the prevalence of moderate and severe mental distress among people aged 20 years and above was 18.9% and 10.5% in 2016, and 18.7% and 10.3% in 2019, respectively [13]. Globally, the COVID-19 pandemic has led to a surge in mental health issues in the general population owing to fear of infection, exposure to uncertain information, and stress associated with the economic recession [15, 17, 18]. According to a meta-analysis, the global prevalence of depression during the COVID-19 pandemic was 28% (23–32%) [15]. Although assessment tools have differed across studies, the reported prevalence rates have been similar to the present study (29.3% prevalence of severe mental distress) [15].

Previous studies have reported that the factors that increase psychological vulnerability to pandemics and disasters include female gender, lower socioeconomic status, interpersonal

**Table 1. Characteristics of study participants (N = 1,200).**

| Variable | *n* (%) |
|---|---|
| **Participant's mental distress (K6 score)** | |
| None (0–4) | 559 (46.6) |
| Moderate (5–9) | 289 (24.1) |
| Severe ($\geq$ 10) | 352 (29.3) |
| **Participant's gender** | |
| Male | 613 (51.1) |
| Female | 587 (48.9) |
| **Participant's age (years)** | |
| < 34 | 123 (10.3) |
| 35–39 | 277 (23.1) |
| 40–45 | 332 (27.7) |
| $\geq$ 45 | 468 (39.0) |
| **Participant's job** | |
| Employed/self-employed | 732 (61.0) |
| Part-time | 184 (15.3) |
| Unemployed/housewife/student | 284 (23.7) |
| **Partner's job** | |
| Employed/self-employed | 725 (60.4) |
| Part-time | 217 (18.1) |
| Unemployed/housewife/student | 258 (21.5) |
| **Size of city of residence (population)** | |
| < 50,000 | 143 (11.9) |
| 50,000–100,000 | 162 (13.5) |
| 100,000–300,000 | 299 (24.9) |
| 300,000–500,000 | 196 (16.3) |
| $\geq$ 500,000 | 400 (33.3) |
| **Time spent with the child during daytime (participant)** | |
| All day | 555 (46.2) |
| Half a day | 217 (18.1) |
| Negligible | 428 (35.7) |
| **Time spent with the child during daytime (partner)** | |
| All day | 433 (36.1) |
| Half a day | 234 (19.5) |
| Negligible | 533 (44.4) |
| **Number of children** | |
| 1 | 376 (31.3) |
| 2 | 646 (53.8) |
| $\geq$ 3 | 178 (14.8) |
| **Child's gender** | |
| Male | 632 (52.7) |
| Female | 568 (47.3) |
| **Child's age (years)** | |
| 3–5 | 300 (25.0) |
| 6–12 | 700 (58.3) |
| $\geq$ 13 | 200 (16.7) |
| **Status of nursery/school** | |
| Open | 109 (9.1) |

(*Continued*)

**Table 1.** (Continued)

| Variable | n (%) |
| --- | --- |
| Online classes/open for children in need | 185 (15.4) |
| Closed | 906 (75.5) |
| **Frequency of playing outside** | |
| Almost every day | 179 (14.9) |
| 3–5 days per week | 291 (24.3) |
| 1–2 days per week | 323 (26.9) |
| Almost never | 407 (33.9) |
| **Screen time compared to before outbreak** | |
| None | 162 (13.5) |
| Less | 33 (2.8) |
| Same | 307 (25.6) |
| Double | 363 (30.3) |
| More than triple | 335 (27.9) |

conflicts, frequent use of social media, inaccurate health information, lower resilience, lack of social support, and pre-existing physical and psychological symptoms [16, 17, 19]. As these risk factors were beyond the scope of this study, it was not possible to determine the reasons for increased mental distress in the study population. Nonetheless, it can be assumed that many people had no access to the necessary social support during the state of emergency because of the suspension of public services such as schools, health checkups, and parental classes. Moreover, many caregivers had to work at home while simultaneously caring for their children, which may have increased psychological stress [5]. Further studies are needed to identify vulnerable populations and develop support programs to prevent mental distress and consequent health problems.

**Table 2. Child health issues reported by participants (N = 1,200).**

| Variable | n (%) |
| --- | --- |
| **Child health issues (overall)** | |
| None | 362 (30.2) |
| At least one | 838 (69.8) |
| **Sleep rhythm change** | |
| No | 512 (42.7) |
| Yes | 688 (57.3) |
| **Appetite change** | |
| No | 859 (71.6) |
| Yes | 341 (28.4) |
| **Physical/mental conditions** | |
| No | 1,068 (89.0) |
| Yes | 132 (11.0) |
| **Activity** | |
| No | 1,000 (83.3) |
| Yes | 200 (16.7) |
| **Behavior change** | |
| No | 1,066 (88.8) |
| Yes | 134 (11.2) |

**Table 3. Crude and adjusted odds ratios of child health issues in relation to caregivers' mental status and other covariates (N = 1,200).**

|  | Crude OR | Adjusted OR[1] |
|---|---|---|
| **Participant's mental distress (K6 score)** |  |  |
| None (0–4) | 1.00 | 1.00 |
| Moderate (5–9) | 2.27 (1.64–3.12) | 2.21 (1.56–3.12) |
| Severe ($\geq$ 10) | 2.86 (2.09–3.91) | 3.11 (2.21–4.39) |
| **Time spent with the child during daytime (participant)** |  |  |
| All day | 1.00 | 1.00 |
| Half a day | 0.86 (0.61–1.22) | 0.98 (0.63–1.50) |
| Negligible | 0.66 (0.50–0.87) | 0.90 (0.62–1.33) |
| **Time spent with the child during daytime (partner)** |  |  |
| All day | 1.00 | 1.00 |
| Half a day | 0.85 (0.60–1.20) | 0.90 (0.59–1.39) |
| Negligible | 0.87 (0.66–1.15) | 0.80 (0.54–1.19) |
| **Child's gender** |  |  |
| Male | 1.00 | 1.00 |
| Female | 0.86 (0.67–1.10) | 1.00 (0.76–1.32) |
| **Child's age (years)** |  |  |
| 3–5 | 1.00 | 1.00 |
| 6–12 | 1.60 (1.20–2.12) | 1.14 (0.78–1.66) |
| $\geq$ 13 | 2.00 (1.34–2.98) | 1.30 (0.76–2.22) |
| **Status of nursery/school** |  |  |
| Open | 1.00 | 1.00 |
| Online classes/open for children in need | 1.91 (1.17–3.13) | 1.04 (0.57–1.90) |
| Closed | 1.96 (1.30–2.93) | 1.18 (0.71–1.97) |
| **Frequency of playing outside** |  |  |
| Almost every day | 1.00 | 1.00 |
| 3–5 days per week | 1.17 (0.80–1.72) | 0.85 (0.55–1.32) |
| 1–2 days per week | 1.77 (1.20–2.61) | 1.41 (0.91–2.18) |
| Almost never | 1.90 (1.30–2.76) | 1.09 (0.70–1.70) |
| **Screen time compared with before outbreak** |  |  |
| None | 1.00 | 1.00 |
| Less | 1.32 (0.62–2.81) | 0.93 (0.42–2.09) |
| Same | 1.48 (1.01–2.17) | 1.37 (0.91–2.06) |
| Double | 3.78 (2.55–5.61) | 3.22 (2.10–4.93) |
| More than triple | 6.44 (4.18–9.93) | 5.60 (3.52–8.90) |

OR: odds ratio

[1]Adjusted for all other variables in the table as well as participant's gender, participant's age, participant's job, partner's job, number of children, and size of city of residence.

Among the participants, 69.8% had children with at least one of the health issues in question. The national online survey of children's quality of life and health during the COVID-19 pandemic conducted at the same time as the present study (between April 30 and May 5) reported that 39% of children felt uncomfortable when thinking of COVID-19 and more than 30% had difficulty concentrating or were easily irritable [20]. It is difficult to directly compare the results, as the two studies used differing terminology to assess different aspects of child health. Furthermore, the respondents differed (i.e., answers were provided by caregivers or children). However, the two studies suggested that children commonly experienced stress

**Table 4. Adjusted odds ratios of child health issues by subdomains (*N* = 1,200).**

| | Sleep rhythm change | Appetite change | Physical/mental conditions | Activity | Behavior change |
|---|---|---|---|---|---|
| **Participant's mental distress (K6 score)** | | | | | |
| None (0–4) | 1.00 | 1.00 | 1.00 | 1.00 | 1.00 |
| Moderate (5–9) | 2.13 (1.56–2.91) | 1.99 (1.43–2.76) | 1.73 (1.02–2.93) | 2.23 (1.50–3.31) | 2.48 (1.48–4.16) |
| Severe ($\geq$ 10) | 2.65 (1.96–3.58) | 1.79 (1.31–2.47) | 3.44 (2.16–5.46) | 1.69 (1.14–2.52) | 3.84 (2.39–6.17) |
| **Child's age (years)** | | | | | |
| 3–5 | 1.00 | 1.00 | 1.00 | 1.00 | 1.00 |
| 6–12 | 1.14 (0.80–1.61) | 0.85 (0.58–1.25) | 1.05 (0.57–1.93) | 0.98 (0.60–1.59) | 1.04 (0.60–1.82) |
| $\geq$ 13 | 1.37 (0.84–2.22) | 1.00 (0.60–1.67) | 1.43 (0.68–3.02) | 0.93 (0.50–1.73) | 0.74 (0.35–1.59) |
| **Status of nursery/school** | | | | | |
| Open | 1.00 | 1.00 | 1.00 | 1.00 | 1.00 |
| Online classes/open for children in need | 1.48 (0.83–2.63) | 1.65 (0.88–3.08) | 3.09 (1.12–8.57) | 1.46 (0.63–3.38) | 1.19 (0.50–2.85) |
| Closed | 1.20 (0.73–1.96) | 1.15 (0.66–1.98) | 1.45 (0.56–3.77) | 1.23 (0.58–2.58) | 0.73 (0.33–1.59) |
| **Frequency of playing outside** | | | | | |
| Almost every day | 1.00 | 1.00 | 1.00 | 1.00 | 1.00 |
| 3–5 days per week | 0.91 (0.60–1.36) | 0.88 (0.57–1.38) | 2.17 (0.84–5.59) | 3.06 (1.48–6.34) | 1.56 (0.79–3.07) |
| 1–2 days per week | 1.53 (1.02–2.29) | 0.87 (0.56–1.36) | 3.92 (1.58–9.72) | 2.58 (1.24–5.37) | 1.13 (0.57–2.25) |
| Almost never | 1.37 (0.91–2.06) | 0.91 (0.58–1.41) | 4.50 (1.82–11.2) | 4.23 (2.07–8.66) | 1.29 (0.65–2.54) |
| **Screen time compared to before outbreak** | | | | | |
| None | 1.00 | 1.00 | 1.00 | 1.00 | 1.00 |
| Less | 1.06 (0.48–2.36) | 1.41 (0.54–3.71) | 1.32 (0.23–7.56) | 0.68 (0.14–3.27) | 1.21 (0.30–4.95) |
| Same | 1.14 (0.76–1.72) | 1.43 (0.84–2.42) | 2.64 (0.97–7.19) | 1.30 (0.64–2.64) | 0.91 (0.39–2.14) |
| Double | 2.16 (1.44–3.26) | 2.85 (1.71–4.76) | 3.41 (1.28–9.06) | 2.15 (1.09–4.22) | 2.11 (0.97–4.59) |
| More than triple | 3.07 (2.01–4.68) | 3.57 (2.14–5.96) | 4.93 (1.86–13.0) | 3.94 (2.04–7.63) | 3.50 (1.64–7.47) |

OR: odds ratio

Adjusted for all other variables in the table as well as participant's gender, participant's age, participant's job, partner's job, participant's time spent with the child during daytime, partner's time spent with the child during daytime, number of children, and size of city of residence.

during the pandemic and presented unpleasant reactions such as anxiety, irritability, and difficulty in sleep. It is important to follow up on whether those conditions have remained, increased, or decreased with the easing of COVID-19-related restrictions, and to determine what factors can mitigate adverse child health conditions. The present study showed that mental distress among caregivers was associated with an increased risk of child health issues. This result is consistent with previous studies reporting a relationship between parental mental health problems and behavioral and physiological problems in their offspring [21–24]. It is known that children and adolescents are particularly vulnerable to stress in the aftermath of natural disasters, owing to their cognitive and psychological immaturity [25]; therefore, they need adult support to adjust to their changed circumstances. In this context, family discordance shortly after a natural disaster could lead to poor mental health and disruptive behavior among children [26]. In particular, the closure of schools and suspension of public health services increased the importance of family functioning during the pandemic. However, given the increased stress of caregivers and the strong association between caregivers' mental distress and child health issues, it would be difficult to rely solely on parents to support and care for children. The development of measures to support isolated families and children during the pandemic is very important.

Another remarkable finding was that in 58.2% of children, screen time was more than double compared to before the pandemic. Although the advantages of using multimedia devices

during the pandemic have been acknowledged [27], increased screen time in the present study may have been associated with recreational purposes, given that only 15.4% of children had access to online classes. This increase in recreational usage might be a result of the disturbance in daily routine because of the closure of nurseries/schools and limited opportunities for outdoor play. This may also suggest limited parental disciplinary ability with regard to ensuring that children adhered to their regular routine during the period of school closure. This increase was associated with a rise in overall child health issues and problems related to sleep, appetite, physical/mental condition, activity, and behavior, independent of the caregiver's mental status. The negative associations between excessive screen time and sleep and mental health problems have also been reported in previous studies [28, 29]. As heavy usage is potentially detrimental to children's physical and psychological health, social connectedness, and academic performance [30], it is necessary to limit children's screen time to help them adhere to routines, engage in healthy activities, and communicate with others.

The present study had several limitations. Owing to the cross-sectional design, the possibility of reverse causation between caregivers' mental health and child health issues cannot be eliminated. In addition, it was difficult to determine whether caregivers' mental distress existed before the pandemic or resulted from it. Data on chronological trends in child health issues were not available either. However, given the stable prevalence of mental distress in the 2016 and 2019 surveys, it would be reasonable to consider that the pandemic influenced the increased prevalence of mental distress among caregivers in the present study. Future studies must compare changes in child health issues before and after the pandemic and determine how long the issues persisted after the lifting of the state of emergency. Additionally, the present study did not consider social factors such as family income and participants' physical and psychological background, which have been reported to be risk factors for mental distress during the COVID-19 pandemic [15, 16]. Further studies considering these factors will be necessary to identify vulnerable populations and to provide customized support for them. Finally, as the examination of child health status was not based on self-reports, caregivers' mental status may have influenced their responses [31]. Therefore, the reported prevalence of child health issues might not reflect the real situation. In addition, the questionnaire did not reflect children's perspectives. Future studies should address the real situation and needs of children by employing self-reports; this information is essential for the provision of effective child-centered support.

## Conclusion

This study is the first to investigate mental health issues in caregivers and their association with child health during the COVID-19 outbreak in Japan. The results showed a tremendous increase in mental distress among caregivers and a strong association between caregivers' mental distress and child health issues. These findings highlight the difficulties faced by caregivers in supporting children. They also suggest the importance of social support for families with children to protect their physical and psychological health during the outbreak. However, both formal and informal social support for children and their families during the state of emergency was quite limited in Japan and caregivers did not have much assistance in caring for their children. There is an urgent need for further research to identify vulnerable populations and children's needs to develop feasible and sustainable social support initiatives specific to infectious outbreaks.

## Supporting information

**S1 File. Original survey questionnaire (Japanese).**
(DOCX)

**S2 File. Survey questionnaire translated in English.**
(DOCX)

**S3 File. Minimal anonymized data.**
(XLSX)

## Acknowledgments

The data collection was supported by the Nippon Research Center.

## Author Contributions

**Conceptualization:** Sayaka Horiuchi, Zentaro Yamagata.

**Formal analysis:** Sayaka Horiuchi.

**Funding acquisition:** Zentaro Yamagata.

**Investigation:** Sayaka Horiuchi, Ryoji Shinohara, Zentaro Yamagata.

**Methodology:** Sayaka Horiuchi, Zentaro Yamagata.

**Project administration:** Sanae Otawa, Zentaro Yamagata.

**Resources:** Yuka Akiyama, Tadao Ooka, Reiji Kojima, Hiroshi Yokomichi, Zentaro Yamagata.

**Supervision:** Ryoji Shinohara, Zentaro Yamagata.

**Writing – original draft:** Sayaka Horiuchi.

**Writing – review & editing:** Sayaka Horiuchi, Ryoji Shinohara, Sanae Otawa, Yuka Akiyama, Tadao Ooka, Reiji Kojima, Hiroshi Yokomichi, Kunio Miyake, Zentaro Yamagata.

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
