## [Decision Letter · Decision Letter 0]

29 Oct 2020

PONE-D-20-28599

Caregivers’ mental distress and child health during the COVID-19 outbreak in Japan

PLOS ONE

Dear Dr. Horiuchi,

Thank you for submitting your manuscript to PLOS ONE. After careful consideration, we feel that it has merit but does not fully meet PLOS ONE’s publication criteria as it currently stands. Therefore, we invite you to submit a revised version of the manuscript that addresses the points raised during the review process.

The two reviewers addressed several major and minor concerns about your manuscript. Please revise your manuscript carefully.

We look forward to receiving your revised manuscript.

Kind regards,

Kenji Hashimoto, PhD

Academic Editor

PLOS ONE

Journal Requirements:

2. Please include additional information regarding the survey or questionnaire used in the study and ensure that you have provided sufficient details that others could replicate the analyses. For instance, if you developed a questionnaire as part of this study and it is not under a copyright more restrictive than CC-BY, please include a copy, in both the original language and English, as Supporting Information.  If the original language is written in non-Latin characters, for example Amharic, Chinese, or Korean, please use a file format that ensures these characters are visible.

3. Please state whether you validated the questionnaire prior to testing on study participants. Please provide details regarding the validation group within the methods section.

4.We note that you have indicated that data from this study are available upon request. PLOS only allows data to be available upon request if there are legal or ethical restrictions on sharing data publicly. For information on unacceptable data access restrictions, please see http://journals.plos.org/plosone/s/data-availability#loc-unacceptable-data-access-restrictions.

Reviewers' comments:

Reviewer's Responses to Questions

**Comments to the Author**

1. Is the manuscript technically sound, and do the data support the conclusions?

Reviewer #1: Partly

Reviewer #2: Yes

2. Has the statistical analysis been performed appropriately and rigorously? 

Reviewer #1: No

Reviewer #2: Yes

3. Have the authors made all data underlying the findings in their manuscript fully available?

Reviewer #1: Yes

Reviewer #2: No

4. Is the manuscript presented in an intelligible fashion and written in standard English?

Reviewer #1: Yes

Reviewer #2: Yes

5. Review Comments to the Author

Reviewer #1: In this population-based epidemiological study, the authors aimed to clarify the physical and mental conditions of children during the coronavirus disease 2019 pandemic and consequent social distancing concerning the mental condition of their caregivers. The present study contributed to a more clear understanding of the influrence of the COVID-19 on caregivers’ mental status and child health issues.

I wish the authors can answer the following questions：

1. Please describe the process of questionnaire collection in detail. What kind of family registered ‘the Nippon Research Center’ before. How many people have received the invitation to participate? What is the response rate? Were all the questionnaires valid?

2. In your result, more than 1/3 caregivers can not spend much time with their children. Maybe the child's health problem is due to the parents' lack of companionship, but not the caregivers’ mental status. How to explain that they don't get along with their children frequently and affect their children's health at the same time.

3. Is it possible that some families have two or more children? Does this affect the result? Would they fill the questionnaire twice or more times？

4. The conclusion in the abstract is ambiguous. ‘The results highlight the infeasibility of parents being solely responsible for the care and support of children.’ But the data did not provide any relevant information to prove the children were only taken care of by parents.

5. Self-reported might be the only way to investigate in the period of COVID-19. The K-6 scale is relatively simple. I suggest the authors use the scale as a measurement but not as a definition of ‘depression’.

6. In the discussion, in 58.2% of children, screen time was more than double compared to before the pandemic . But in result 'in the case of 15.4% of the participants, the children had access to either online classes or special offline classes.' This indirectly means that many Internet users may be recreational. This should be discussed in detail.

7. In result Table 3 and 4, only 'OR' can be seen. Why there were no 'p'. values. How do we know if it's significant?

Reviewer #2: 1. Line 39. “infeasibility” is an intense word to use without being to state causation

2. Formatting is inconsistent throughout the manuscript (e.g., Odds ratios,

3. Need references for government declarations (i.e., line 50, 52, & 70)

4. Mental issues seems inappropriate (mental disorders or distress would be more appropriate (i.e., line 77 & 79)

5. What is the reliability of the Kessler Psychological Distress Scale-6? What was the alpha for this study?

6. Line 90 references a questionnaire developed by the authors but does not describe the questionnaire

7. It is unclear what the authors are referring to with the “exposure” section. What is an exposure?

8. Is it typical to not include questions due to concern about sensitivity? Seems like a big oversight to not ask about SES, education, or medical history

9. Line 128-129 “as age and gender are important factors…” needs a reference

10. Line 143 I believe it should be “there was no missing data”

11. Table 1 described the K6 as indicating participant’s mental status and this also seems inappropriate (mental distress or wellbeing would be more appropriate)

12. Line 187 this sentence is unclear and should b e re-written

13. Line 192 what vulnerability are you describing? (e.g., mental distress)

14. A lot of factors were missed described as beyond the scope of this study which seems to limit the utility of this study

15. Line 209 althought it may be difficult to compare with incommensurable measures I believe some comparison of the results could be further discussed.

16. Line 238 “owing” should be replaced with another word (e.g., due)

17. Line 243 seems to be a large assumption that the pandemic worsened caregivers mental health due to the results of this study as there is no data of their mental health prior to the pandemic

18. Formatting of references must be consistent (e.g., hyperlink)

6. PLOS authors have the option to publish the peer review history of their article (what does this mean?). If published, this will include your full peer review and any attached files.

Reviewer #1: No

Reviewer #2: **Yes: **Ashley A. Balsom

---

## [Author Response · Author response to Decision Letter 0]

11 Nov 2020

Editor: We really appreciate your consideration for our study and valuable comments to improve it. Our response to each of the comments are as follows. 

Reply: We carefully have checked and followed the instructions to meet PLOS ONE’s style requirements.

2. Please include additional information regarding the survey or questionnaire used in the study and ensure that you have provided sufficient details that others could replicate the analyses. For instance, if you developed a questionnaire as part of this study and it is not under a copyright more restrictive than CC-BY, please include a copy, in both the original language and English, as Supporting Information. If the original language is written in non-Latin characters, for example Amharic, Chinese, or Korean, please use a file format that ensures these characters are visible.

Reply: We have attached the original questionnaire in Japanese (original) and English translation as supporting information.

“S1 File. Original survey questionnaire (Japanese) 

S2 File. Survey questionnaire translated in English”

3. Please state whether you validated the questionnaire prior to testing on study participants. Please provide details regarding the validation group within the methods section.

Reply: We did not validate the questionnaire prior to the survey. The Japanese version of the Kessler Psychological Distress Scale-6 (K6), which was included in the questionnaire, has proven valid. We have added details in the methods section as follows:

(lines 86-90) “The questionnaire was developed by the authors. No validation test was performed prior to the survey. However, the Japanese version of the Kessler Psychological Distress Scale-6 (K6), which was included in the questionnaire, has proven validity in the context of mental health screening.9 The Cronbach’s alpha coefficient of the K6 score in this study was 0.93.”

4.We note that you have indicated that data from this study are available upon request. PLOS only allows data to be available upon request if there are legal or ethical restrictions on sharing data publicly. For information on unacceptable data access restrictions, please see http://journals.plos.org/plosone/s/data-availability#loc-unacceptable-data-access-restrictions.

Reply: There are no legal or ethical restrictions on sharing data, and we would like to share the data publicly.

Reply: We have attached the minimal anonymized data set as supporting information. Consent for publication of raw data not obtained but dataset is fully anonymous in a manner that can easily be verified by any user of the dataset. Publication of the dataset clearly and obviously presents minimal risk to confidentiality of study participants. This statement has been included in the revised cover letter.

1. Thank you for including your ethics statement on the online submission form: "This study was approved by the Ethics Review Board of the University of Yamanashi (approval number: 2259). Informed consent was obtained via online.". 

To help ensure that the wording of your manuscript is suitable for publication, would you please also add this statement at the beginning of the Methods section of your manuscript file.

Reply: We appreciate your suggestion and have included the ethics statement in the Methods section.

(line 67-68) “This study was approved by the Ethics Review Board of the University of Yamanashi (approval number: 2259). Informed consent was obtained via online.”

 

Reviewer #1: We appreciate your valuable comments to improve our study, and accordingly have attempted to address all of your concerns, as below.

Reviewer #1: In this population-based epidemiological study, the authors aimed to clarify the physical and mental conditions of children during the coronavirus disease 2019 pandemic and consequent social distancing concerning the mental condition of their caregivers. The present study contributed to a more clear understanding of the influence of the COVID-19 on caregivers’ mental status and child health issues.

I wish the authors can answer the following questions：

1. Please describe the process of questionnaire collection in detail. What kind of family registered ‘the Nippon Research Center’ before. How many people have received the invitation to participate? What is the response rate? Were all the questionnaires valid?

Reply: We appreciate your valuable suggestions. Registration to the Nippon Research Center is done on voluntary basis through online affiliate advertising. The Nippon research center requested 2792 people to answer the questionnaire, and 1200 people who answered the entire questionnaire were included in this study. While the questionnaire that was developed by the authors were not validated prior to the survey, the Japanese version of the Kessler Psychological Distress Scale-6 (K6), which was included in the questionnaire, has proven valid. We have added explanation in the method and result sections as follows.

(lines 74-78) “The target population was caregivers of children aged 3–14 years who had voluntarily registered with the Nippon Research Center as monitors for web-based surveys in response to online affiliate advertising. Eligible individuals were asked to participate in the survey through the center and only those who answered screening questions and provided their consent on the website answered the entire questionnaire.”

(line 86-90) “The questionnaire was developed by the authors. No validation test was performed prior to the survey. However, the Japanese version of the Kessler Psychological Distress Scale-6 (K6), which was included in the questionnaire, has proven validity in the context of mental health screening.9 The Cronbach’s alpha coefficient of the K6 score in this study was 0.93.”

(lines 101-103) “Data collection was commissioned by the Nippon Research Center. If participants had more than one child, they were asked to restrict their answers to one child, who was randomly selected by the system.”

(lines 148-151) “The Nippon research center requested a total of 2792 people to answer the questionnaire. Among 1748 people who answered screening questions, 548 people were removed because they did not meet eligibility criteria (caregivers of children aged 3–14 years).”

2. In your result, more than 1/3 caregivers cannot spend much time with their children. Maybe the child's health problem is due to the parents' lack of companionship, but not the caregivers’ mental status. How to explain that they don't get along with their children frequently and affect their children's health at the same time.

Reply: In fact, respondent’s time spent with the child during daytime was inversely associated with child health issue in univariate analysis and the association disappeared in multivariate analysis. Partner’s time spent with the child during daytime was not associated with child health issue either in univariate or multivariate analyses. Therefore, we included time spent with the child in multivariate analyses but did not present its effect estimates in Table 3. We now have included the results (effect estimates of time spent with the child) in Table 3 in the revised manuscript. Caregivers’ mental distress was still associated with child health issue after adjusting for respondent’s and partner’s time spent with the child during daytime.

3. Is it possible that some families have two or more children? Does this affect the result? Would they fill the questionnaire twice or more times？

Reply: As you pointed out, about 70% of the respondents had more than two children. The respondent answered the questionnaire only once regarding one of their children who was randomly selected by the system. The choice of the child was explained as follows in the method section. 

(lines 101-103) “If participants had more than one child, they were asked to restrict their answers to one child, who was randomly selected by the system.”

We have added the number of children in the multivariate analysis (Tables 3 and 4). The number of children was not associated with child health issue. It did not influence on the association between child health issue and respondent’s mental status (K6 score) either. 

4. The conclusion in the abstract is ambiguous. ‘The results highlight the infeasibility of parents being solely responsible for the care and support of children.’ But the data did not provide any relevant information to prove the children were only taken care of by parents.

Reply: We greatly appreciate your points to improve our manuscript. We have revised the part as follows:

(lines 29-31) “The results highlight parents’ psychological stress during the pandemic, suggesting the need for adequate parenting support.”

5. Self-reported might be the only way to investigate in the period of COVID-19. The K-6 scale is relatively simple. I suggest the authors use the scale as a measurement but not as a definition of ‘depression’.

Reply: We agree on your point. We have deleted depression from the label of the K6 category. As the K6 has been utilized as a screening tool of depression and anxiety disorder in Japan, we would like to add the following explanation in the method section for readers.

(lines 94-98) “The properties of the Japanese version are comparable to those of the original (the area under the receiver operating characteristic curve was 0.94 (95% confidence interval [CI] = 0.88 to 0.99),9 and this tool has been used to screen for depression and anxiety disorders in the workplace and in the Comprehensive Survey of Living Conditions in Japan.13”

Reference 13. Ministry of Health, Labour and Welfare. Comprehensive Survey of Living Conditions [cited 2020 Jul 28]. Available from: https://www.mhlw.go.jp/toukei/saikin/hw/k-tyosa/k-tyosa16/dl/04.pdf

6. In the discussion, in 58.2% of children, screen time was more than double compared to before the pandemic. But in result 'in the case of 15.4% of the participants, the children had access to either online classes or special offline classes.' This indirectly means that many Internet users may be recreational. This should be discussed in detail.

Reply: We appreciate your insightful comment. We would like to discuss this point in detail as follows.

(lines 247-254) “Although the advantages of using multimedia devices during the pandemic have been acknowledged,28 increased screen time in the present study may have been associated with recreational purposes, given that only 15.4% of children had access to online classes. This increase in recreational usage might be a result of the disturbance in daily routine because of the closure of nurseries/schools and limited opportunities for outdoor play. This may also suggest limited parental disciplinary ability with regard to ensuring that children adhered to their regular routine during the period of school closure.”

7. In result Table 3 and 4, only 'OR' can be seen. Why there were no 'p'. values. How do we know if it's significant?

Reply: We did not include p-values as confidence intervals that do not cross 1 also suggest statistically significant association.

 

Reviewer #2: We appreciate your valuable comments to improve our study, and accordingly have attempted to address all of your concerns, as below.

1. Line 39. “infeasibility” is an intense word to use without being to state causation

Reply: We agree on your point. We have revised the part as follows:

(lines 29-31) “The results highlight parents’ psychological stress during the pandemic, suggesting the need for adequate parenting support.”

2. Formatting is inconsistent throughout the manuscript (e.g., Odds ratios,

Reply: We appreciate your suggestion. We would like to use “OR” throughout the main part and use “odds ratio” in the abstract to avoid abbreviation. We hope that it would be acceptable.

3. Need references for government declarations (i.e., line 50, 52, & 70)

Reply: We greatly appreciate your suggestion and have added government website in the reference list.

“6. Ministry of Education, Culture, Sports, Science and Technology. Information on MEXT’s measures against COVID-19 [cited 2020 October 31]. Available from: https://www.mext.go.jp/en/mext_00006.html

7. Prime Minister of Japan and His Cabinet. [COVID-19] Declaration of a State of Emergency in response to the Novel Coronavirus Disease (April 7) [cited 2020 October 31]. Available from: https://japan.kantei.go.jp/ongoingtopics/_00018.html

8. Prime Minister of Japan and His Cabinet. [COVID-19] The Declaration of the Lifting of the State of Emergency in Response to the Novel Coronavirus Disease [cited 2020 October 31]. Available from: https://japan.kantei.go.jp/ongoingtopics/_00027.html”

4. Mental issues seems inappropriate (mental disorders or distress would be more appropriate (i.e., line 77 & 79)

Reply: We agree on your suggestion and have revised the sentence as follows accordingly:

(lines 79-83) “The sample size was calculated to detect a 20% absolute difference in the percentage of children with any physical or mental health problems between caregivers with and without mental distress based on reports of a 40–50% prevalence of mental distress in the general population during the pandemic.”

5. What is the reliability of the Kessler Psychological Distress Scale-6? What was the alpha for this study?

Reply: The areas under receiver operating characteristic curves (AUC) was reported as 0.94 (95% confidence interval = 0.88 to 0.99) for K6 by Furukawa et al. (The performance of the Japanese version of the K6 and K10 in the World Mental Health Survey Japan, 2008). The scale reliability coefficient was 0.93 in the present study. We have added explanation as follows:

(lines 89-90) “The Cronbach’s alpha coefficient of the K6 score in this study was 0.93.”

(lines 94-98) “The properties of the Japanese version are comparable to those of the original (the area under the receiver operating characteristic curve was 0.94 (95% confidence interval [CI] = 0.88 to 0.99),9 and this tool has been used to screen for depression and anxiety disorders in the workplace and in the Comprehensive Survey of Living Conditions in Japan.13”

6. Line 90 references a questionnaire developed by the authors but does not describe the questionnaire

Reply: The contents of the questionnaire were described in lines 89-98. We have added the questionnaire (Japanese original version and English translation) in supporting information.

(lines 91-100) “Participant-related questions included job type, time spent with the child, COVID-19-related concerns, and mental health status. Mental health status was assessed using the Japanese version of the K6.9,10 The K6 has demonstrated excellent internal consistency and reliability11 and is widely used in epidemiological studies.12 The properties of the Japanese version are comparable to those of the original (the area under the receiver operating characteristic curve was 0.94 (95% confidence interval [CI] = 0.88 to 0.99),9 and this tool has been used to screen for depression and anxiety disorders in the workplace and in the Comprehensive Survey of Living Conditions in Japan.13 Questions about the child included whether the school/nursery was open, frequency of playing outside, screen time, and health condition as perceived by the participant.”

“Supporting information

S1 File. Original survey questionnaire (Japanese) 

S2 File. Survey questionnaire translated in English”

7. It is unclear what the authors are referring to with the “exposure” section. What is an exposure?

Reply: The main exposure was caregivers’ mental status measured using the Japanese version of the K6.

8. Is it typical to not include questions due to concern about sensitivity? Seems like a big oversight to not ask about SES, education, or medical history

Reply: We agree that SES, educational status and medical history are important factors to consider in analyzing effects of mental distress of caregivers on child health issues during the pandemic. The survey was conducted in quite urgent manner to understand the situation of physical and psychological status and concerns of parents and children during the state of emergency period. Due to concern that some people might refrain answering the survey due to private questions, we had to omit some questions to maintain easiness to answer and raise the response rate. Nonetheless, we still asked about job type of respondents and their partners. We have included the job type in the analyses, which did not alter our conclusions. We would like to try to collect more data on SES and medical history in follow-up surveys.

(lines 126-133) “The family’s socioeconomic status has previously been reported to be a risk factor for child mental health problems,5,15,16 and the questionnaire therefore asked about the respondents’ job type and their partners. We categorized the job type into employed/self-employed, part-time, and unemployed/housewife/student. However other potential risk factors for child mental health problems such as educational attainment15,16 and past medical history16 were not included in the questionnaire, owing to the sensitivity of these topics and with a view to increasing the response rate.”

9. Line 128-129 “as age and gender are important factors…” needs a reference

Reply: References are attached as follows:

(lines 143-145) “….as age and gender are important factors in determining children’s vulnerability to environmental change, such as the COVID-19 outbreak, according to previous studies.15,16” 

15. Luo M, Guo L, Yu M, Jiang W, Wang H. The psychological and mental impact of coronavirus disease 2019 (COVID-19) on medical staff and general public-a systematic review and meta-analysis. Psych Res 2020; 291:113190.

16. Vindegaard N, Eriksen Benros M. COVID-19 pandemic and mental health consequences: systematic review of the current evidence. Brain Behav Immun 2020. doi:10.1016/j.bbi.2020.05.048. 

10. Line 143 I believe it should be “there was no missing data”

Reply: We appreciate your suggestion. An English editor have checked it again and confirmed that “Data” is always used in the plural. We would like to keep the original sentence: “There were no missing data.”

11. Table 1 described the K6 as indicating participant’s mental status and this also seems inappropriate (mental distress or wellbeing would be more appropriate)

Reply: We agree on your point and have revised the wording accordingly. 

“Participant’s mental distress (K6 score): None (0–4), Moderate (5–9), Severe (≥ 10)”

12. Line 187 this sentence is unclear and should b e re-written

Reply: We have revised the sentence as follows. We hope this is readable and acceptable.

(lines 201-203) “Globally, the COVID-19 pandemic has led to a surge in mental health issues in the general population owing to fear of infection, exposure to uncertain information, and stress associated with the economic recession.”

13. Line 192 what vulnerability are you describing? (e.g., mental distress)

Reply: We appreciate your suggestion and have revised the sentence for more clarity as follows:

(lines 208-209) “Previous studies have reported that the factors that increase psychological vulnerability to pandemics and disasters”

14. A lot of factors were missed described as beyond the scope of this study which seems to limit the utility of this study

Reply: We appreciate your comments. The primary objective of the present study was to investigate the prevalence of mental distress among parents, the prevalence of health issues among children and the relationship between them. Therefore, we did not collect much data on characteristics of parents that can be risk factors of mental distress. Therefore, we could not consider what types of parents and households are at risk of mental distress during the outbreak in the present study. We believe that this would not influence the internal validity but the generalizability of our study. Further studies will be needed to understand more about those who are at risk of mental distress and in need of social support. It is explained in the limitation part as follows. 

(lines 271-275) “Additionally, the present study did not consider social factors such as family income and participants’ physical and psychological background, which have been reported to be risk factors for mental distress during the COVID-19 pandemic.15,16 Further studies considering these factors will be necessary to identify vulnerable populations and to provide customized support for them.”

15. Line 209 althought it may be difficult to compare with incommensurable measures I believe some comparison of the results could be further discussed.

Reply: We appreciate your suggestion. While it is difficult to directly compare the prevalence of child health issues between the two studies, we have added consideration from the two studies as follows.

(lines 224-229) “It is difficult to directly compare the results, as the two studies used differing terminology to assess different aspects of child health. Furthermore, the respondents differed (i.e., answers were provided by caregivers or children). However, the two studies suggested that children commonly experienced stress during the pandemic and presented unpleasant reactions such as anxiety, irritability, and difficulty in sleep.”

17. Line 238 “owing” should be replaced with another word (e.g., due)

Reply: We appreciate your suggestion. An English editor have checked it again and confirmed that “owing to” is the best fit in this sentence. We would like to keep the original sentence.

(lines 262-264) “Owing to the cross-sectional design, the possibility of reverse causation between caregivers’ mental health and child health issues cannot be eliminated.”

17. Line 243 seems to be a large assumption that the pandemic worsened caregivers mental health due to the results of this study as there is no data of their mental health prior to the pandemic

Reply: We understand your point. The prevalence of the mental distress in the present study is tremendously higher compared to the national survey that evaluate the prevalence of mental distress using K6 every three years, which reported about 10% of severe mental distress constantly every time in previous 6 years. We consider that the dramatic change in the prevalence of mental distress between the time of survey and the present study would suggest the influence of the pandemic on mental health. We have added the results of 2019 national survey that are newly published. We have also revised the sentence as follows:

(lines 197-201) “The present study revealed that the number of caregivers with moderate to severe mental distress during the state of emergency was more than double the number observed in national surveys, wherein the prevalence of moderate and severe mental distress among people aged 20 years and above was 18.9% and 10.5% in 2016, and 18.7% and 10.3% in 2019, respectively.”

(lines 266-269) “However, given the stable prevalence of mental distress in the 2016 and 2019 surveys, it would be reasonable to consider that the pandemic influenced the increased prevalence of mental distress among caregivers in the present study.”

18. Formatting of references must be consistent (e.g., hyperlink)

Reply: We have reviewed the reference list and removed the hyperlink.

---

## [Decision Letter · Decision Letter 1]

27 Nov 2020

Caregivers’ mental distress and child health during the COVID-19 outbreak in Japan

PONE-D-20-28599R1

Dear Dr. Horiuchi,

We’re pleased to inform you that your manuscript has been judged scientifically suitable for publication and will be formally accepted for publication once it meets all outstanding technical requirements.

Kind regards,

Kenji Hashimoto, PhD

Section Editor

PLOS ONE

Additional Editor Comments (optional):

Reviewers' comments:

Reviewer's Responses to Questions

**Comments to the Author**

1. If the authors have adequately addressed your comments raised in a previous round of review and you feel that this manuscript is now acceptable for publication, you may indicate that here to bypass the “Comments to the Author” section, enter your conflict of interest statement in the “Confidential to Editor” section, and submit your "Accept" recommendation.

Reviewer #1: All comments have been addressed

Reviewer #2: All comments have been addressed

2. Is the manuscript technically sound, and do the data support the conclusions?

Reviewer #1: Yes

Reviewer #2: Yes

3. Has the statistical analysis been performed appropriately and rigorously? 

Reviewer #1: N/A

Reviewer #2: Yes

4. Have the authors made all data underlying the findings in their manuscript fully available?

Reviewer #1: Yes

Reviewer #2: Yes

5. Is the manuscript presented in an intelligible fashion and written in standard English?

Reviewer #1: Yes

Reviewer #2: Yes

6. Review Comments to the Author

Reviewer #1: (No Response)

Reviewer #2: I believe that the authors have thoughtfully and thoroughly addressed previous reviewer comments and am recommending that the manuscript be accepted.

7. PLOS authors have the option to publish the peer review history of their article (what does this mean?). If published, this will include your full peer review and any attached files.

Reviewer #1: No

Reviewer #2: No

---

## [Editor Report · Acceptance letter]

2 Dec 2020

PONE-D-20-28599R1 

Caregivers’ mental distress and child health during the COVID-19 outbreak in Japan 

Dear Dr. Horiuchi:

I'm pleased to inform you that your manuscript has been deemed suitable for publication in PLOS ONE. Congratulations! Your manuscript is now with our production department. 

Kind regards, 

on behalf of

Prof. Kenji Hashimoto 

Section Editor

PLOS ONE